# Development of a Holistic In Vitro Cell-Free Approach to Determine the Redox Bioactivity of Agricultural Products

**DOI:** 10.3390/ijms242216447

**Published:** 2023-11-17

**Authors:** Zoi Skaperda, Fotios Tekos, Periklis Vardakas, Paraskevi-Maria Nechalioti, Maria Kourti, Anastasia Patouna, Sotiria Makri, Maria Gkasdrogka, Demetrios Kouretas

**Affiliations:** Department of Biochemistry and Biotechnology, University of Thessaly, 41500 Larissa, Greece; zoiskap94@gmail.com (Z.S.); ftekos@uth.gr (F.T.); periklis_vardakas94@hotmail.com (P.V.); mariakourti@uth.gr (M.K.); anastasia.pat93@hotmail.com (A.P.); sotirina_m@hotmail.com (S.M.); mariagkasd@gmail.com (M.G.)

**Keywords:** agricultural products, methodology, cell-free assays, antioxidant capacity, bioactivity

## Abstract

In recent years, there has been a strong consumer demand for food products that provide nutritional benefits to human health. Therefore, the assessment of the biological activity is considered as an important parameter for the promotion of high-quality food products. Herein, we introduce a novel methodology comprising a complete set of in vitro cell-free screening techniques for the evaluation of the bioactivity of various food products on the basis of their antioxidant capacity. These assays examine the free radical scavenging activities, the reducing properties, and the protective ability against oxidative damage to biomolecules. The adoption of the proposed battery of antioxidant assays is anticipated to contribute to the holistic characterization of the bioactivity of the food product under examination. Consumer motivations and expectations with respect to nutritious food products with bio-functional properties drive the global food market toward food certification. Therefore, the development and application of scientific methodologies that examine the quality characteristics of food products could increase consumers’ trust and promote their beneficial properties for human health.

## 1. Introduction

The term “stress” was firstly introduced in biological sciences as a non-specific response of an organism against several exogenous factors. In 1956, Hans Selye was the first scientist to refer to this term in the context of human physiology, describing generic signs and symptoms that are responsible for several pathologies and illnesses. It was only in 1970 when the term “oxidative stress” was mentioned by Paniker [1], indicating the detrimental effects of oxidizing agents, such as hydrogen peroxide (H_2_O_2_). Over the years, this concept has been redefined, and nowadays, oxidative stress is described as “an imbalance between oxidants and antioxidants in favor of the oxidants, leading to a disruption of redox signaling and control and/or molecular damage” [2]. The inability of an organism to regulate free radicals is a critical step toward the induction of oxidative stress, which is counterbalanced by the activities of endogenous and exogenous antioxidants. Therefore, the protective role of antioxidants against oxidative stress has been widely investigated. Furthermore, antioxidants have been proposed as additives in the food industry to delay, retard or prevent the development of rancidity or other flavor deterioration in food products due to oxidation. Antioxidants exert their protective effects by scavenging free radicals, chelating metal ions, converting hydroperoxides to non-radical species, absorbing UV radiation or deactivating singlet oxygen [3]. Their mechanism of action depends mainly on the chemical structure of the antioxidant.

Regardless of the mechanism of action, it was supposed that antioxidants “cure” oxidative stress, caused by the overproduction of free radicals or the inability of the endogenous antioxidant mechanisms to effectively neutralize them. Nevertheless, the use of antioxidants in large-scale clinical trials have shown no significant beneficial results. Therefore, “cure” was not suitable to describe their role in oxidative stress, which eventually occurs due to an alteration of the thiol redox state, leading to a disruption of cellular signaling and of physiological function.

Based on the alterations of thiol redox circuits and the up- or down-regulation of reactive oxygen species (ROS), the scientific literature has distinguished oxidative stress in “oxidative eustress” and “oxidative distress” [4,5]. The first is responsible for providing beneficial signal transduction and regulating several biochemical reactions due to low ROS levels, whereas the latter is responsible for disrupting redox signaling and causing molecular damage due to excessive ROS levels. It has to be noted that low oxidative events do not always participate in physiologically positive actions, while high oxidative events do not always exert negative actions, indicating that the complex and fine regulation of ROS generation is not easily discrete. An example of the difficulty to define high and low amounts of oxidants and their relative functions is preconditioning, a mechanism by which small amounts of an oxidant provided to cells at different times and in different amounts result in a resistance to the damage caused by high amounts of oxidants. All of these occurrences, described as stress, could be topographically distinct, since the regulation or damage could be restricted to a specific site, while also occurring at the same time in different sites.

The development of reliable and valid methodologies that evaluate the antioxidant properties of food products is the first step toward the improvement of knowledge regarding their beneficial or detrimental effects for human health. More specifically, in this review, we introduce a holistic approach to determine the bioactivity of various agricultural products, based on a panel of in vitro cell-free assays. This approach will contribute to the creation of added value for food products which were traditionally considered superior to others, however lacking in appropriate scientific data, thus providing to the consumers food products of higher quality.

## 2. The Importance of Quality Assessment in Food Industry

The determination of food authenticity and quality includes the identification of mislabeled products that do not meet the requirements to be characterized as bio-functional [6]. Due to the high production costs in the food industry, in many cases, high-quality foods/ingredients are replaced by similar less expensive kinds, even those of dubious quality [7]. The adoption of undeclared procedures, the food adulteration, and the incorrect declaration of the origin of raw materials or of the production method are some of the common practices for downgrading production costs, also leading to lower food quality [7].

Nowadays, the food quality assessment attracts considerable interest as consumers come into contact with a wide variety of food products on a daily basis, and among the main selection criteria is food certification. Since globalization has permitted the unlimited facile trade of an increasing number of food products, their traceability and certification of quality has become one of the cornerstones of the European Union (EU)’s food safety policy [8]. Therefore, there is a strong trend for the development of tools that will enable the food industry to satisfy the underlining consumer need to be ensured that their food products are of high quality and that they exert beneficial effects to human health based on their bioactivity [7]. Many thinkers of the 21st century consider that consumers can be the critical “revolutionary mass” that will attempt the next historical socio-economic revolution, changing the current structure of the food production model, but mainly restructuring the existing model of food labeling and the health-translational potency of these labels.

For food products protected by geographical indications and traditional specialties and, to be more specific, the Protected Geographical Indications (PGI), Protected Designation of Origin (PDO), and Traditional Specialties Guarantee (TSG), the EU Regulations EC N. 510/2006 [9] and 1151/2012 [10] require several protection measures against mislabeling. Thereupon, Regulation N. 668/2014 implements specific rules for the application of Regulation N. 1151/2012 of the European Parliament and of the Council on quality schemes for agricultural products and foodstuffs [11]. Except for the geographical indications, the EU Commission has established rules, principles, and requirements that farmers need to comply with in order to be certified for organic farming [12]. Depending on each separate Member State, the authorities in charge of the control system for organic production may confer their control competences to one or more public control authorities or delegate control tasks to one or more private control bodies. The Commission must be aware of the list of designated control authorities and approved control bodies in each Member State [13]. Furthermore, all food producers, processors or traders who wish to market their food as organic need to be aware of the registration process with the control agency or body. Yearly inspections and the checking of complying with the rules of organic production are well operated by the control agencies or bodies of each Member State. Although in most cases, the incorporated traceability systems guarantee the geographical origin of food products [9], and also, conforming with the organic farming rules is a well-described requirement by the EU Commission, concerning their retail market, the food industry urgently needs screening methods for the unlabeled foods to provide proofs of their quality in order to educate and inform consumers whether a food product is “good” or “bad” for their health. This “gap” must be filled using certain protocol schemes that shall examine endogenous bioactive properties of the final food products with a homogenized system.

Geographical, climatic, pedological, geological, botanical, and agricultural parameters affect the ratios and patterns of bio-elements in nature, and these variations are incorporated into the plant or animal tissues throughout the food chain and through direct contact with the natural environment. Food products contain hundreds of chemical constituents in varying concentrations and the comprehensive analysis of their bioactivity can reveal unique distinctive patterns. The screening of the biological properties of a wide variety of both lipophilic and hydrophilic antioxidant substances using a rapid and low-cost methodology is a critical step toward establishing the classification of the endogenous bioactivity of foodstuffs. Toward this purpose, in the present contribution, we propose the utilization of a panel of in vitro cell-free assays that evaluate the antioxidant capacity of food products on the basis of their antiradical, reducing, and biomolecule protective properties. To be more specific, the antiradical activities can be determined spectrophotometrically by assessing the free radical scavenging capacity against the synthetic 2,2-diphenyl-1-picrylhydrazyl (DPPH^•^) and 2,2′-azino-bis(3-ethylbenzothiazoline-6-sulfonic acid (ABTS^•+^) free radicals, as well as against the natural superoxide anion (O_2_^•−^) and hydroxyl (OH^•^) radicals. With regard to the reducing properties, they can be determined spectrophotometrically by evaluating the ferric reducing ability using reducing power and ferric reducing antioxidant power (FRAP) assays, as well as the cupric reducing ability via cupric ion reducing antioxidant capacity (CUPRAC) assay. Finally, concerning their ability to protect endogenous biomolecules from oxidation, the spectrophotometric thiobarbituric acid reactive substance (TBARS) assay can be used to assess the capacity of food products to prevent from lipid peroxidation, as well as the electrophoretic plasmid DNA relaxation assay to evaluate the potency of food products to protect DNA from oxidative damage.

## 3. Characterization of the Food Product Quality on the Basis of the Antioxidant Profile

Dietary antioxidants, present in several food products, possess the ability to neutralize the excess of free radicals, which are produced as by-products of normal cell metabolism [14]. Therefore, the consumption of food products that are rich in bioactive compounds with antioxidant properties might protect against the onset and the progression of pathological conditions associated with disturbances of redox homeostasis [14]. In addition to the benefits for human health, the antioxidant profile affects the shelf life and the flavor stability of a food product and protects its ingredients from oxidations, which could lead to quality degradation.

An important issue in the global food industry is the inability of the consumers to recognize food products that are not only a source of primary nourishment, but are also capable of exerting beneficial health effects [15]. Therefore, it is critical to develop a specific scheme of laboratory analyses that could examine the food product quality based on their antioxidant properties and categorize them according to their bioactivity [16].

Up to the present time, several antioxidant assays have been introduced to investigate the antioxidant properties of conventional antioxidants, foodstuffs, dietary supplements, and biological samples (Table 1). In the table below, natural products, such as wines [17], berries [18], honey [19], herbs [20], grape seed extracts [21], and plant extracts [22], have been examined using experimental protocols that investigate their endogenous biological properties, such as antiradical potency, as well as their reducing capacity and DNA protective activity in relation with their antioxidant power. In order to characterize the bioactivity of a food product, in terms of its antioxidant potency, the first and crucial step is the adoption of a battery of reliable and valid antioxidant markers. Herein, we propose the establishment of a network of antioxidant markers with translational potency, which can holistically evaluate the antioxidant properties of food products. The main advantages of the proposed methodology is the simplicity of the experimental procedures, the good within-run and the between-day reproducibility, as well as the detection of both hydrophilic and lipophilic antioxidants.

## 4. Development of a Novel Methodology for the Evaluation of the Antioxidant Capacity of Agri-Food Products

Agricultural products, such as fruits, vegetables, legumes, and grains contain high amounts of phenolic compounds, plant secondary metabolites with strong antioxidant properties that are broadly distributed in higher plants [23,24,25]. In particular, they comprise a vast, diverse class of bioactive phytochemicals with common structural characteristics: the presence of at least one aromatic ring bearing one or more hydroxyl substituents [26]. It is worth noting that the antioxidant activities of phenolic compounds are significantly affected by their chemical structure and, in this context, Bors criteria have been proposed to describe their antioxidant behavior [27]. Firstly, the existence of a catechol group on B-ring (Bors 1) confers an increased stability to the formed radical [28]. Secondly, the combination of a 2,3 double bond and a 4-oxo group on C-ring (Bors 2) enhances the delocalization of electrons [28]. Finally, the existence of hydroxyl groups at positions 3 and 5 in conjunction with a 4-oxo group (Bors 3) facilitates the delocalization of electrons via hydrogen bonding [28].

Protocols that investigate the antioxidant profile of food products in in vitro cell-free systems should ensure comparability and reproducibility between different measurements [29]. The methods proposed herein can be applied for the evaluation of the antioxidant properties of both lipophilic and hydrophilic bioactive compounds in the tested food products, providing a complete set of in vitro cell-free screening techniques. It has been proposed that assays which involve peroxyl radical scavenging should be taken seriously into account, being one of the most dominant inductors of neurodegenerative and other inflammatory diseases [30]. Although 2,2′-azino-bis(3-ethylbenzothiazoline-6-sulfonic acid (ABTS) and 2,2-diphenyl-1-picrylhydrazyl (DPPH) assays rely on slow reactions that are sensitive to ascorbic acid, uric acid, and polyphenols, thus enabling secondary reactions that are likely to occur and yield false-positive results in some cases, they are preferable since they are easy to be conducted and are rapid. In the following sections, we propose the utilization of various cell-free assays for the determination of the bioactivity of food products based on multiple chemical reactions.

In order to compare the results of antioxidant methods between different laboratories, a single unit should be established as a standard that can be used for the comparability of the results. In all assays that are presented below, apart from the reducing power assay, the half maximal inhibitory concentration (IC_50_) represents the concentration of the sample that it is required for the inhibition of the 50% of the corresponding free radical. Therefore, the lower the IC_50_ value, the greater the antioxidant capacity of the sample tested. For the reducing power assay, an absorbance unit of 0.5 (AU_0.5_) value is used, representing the concentration of the tested sample required for the achievement of an absorbance value of 0.5.

### 4.1. DPPH^•^ Radical Scavenging Assay

The DPPH^•^ radical scavenging assay was firstly introduced by Brand-Williams et al. [31] for the determination of the antioxidant capacity of a substance. It represents a rapid, simple, and reliable method for the preliminary assessment of the antioxidant strength of food products that contain mainly lipophilic antioxidant compounds [32]. The assessment is based on the interaction of the antioxidant compounds contained in food products, with the stable, synthetic DPPH^•^ radical. The DPPH^•^ radical is neutralized by receiving an electron or a hydrogen proton as follows:DPPH^•^ _(violet at 517 nm)_ + ArOH → DPPH(H) _(yellow)_ + ArO^•^

The quantification of the organic nitrogen DPPH^•^ radical can be conducted spectrophotometrically at 517 nm. The reaction is based on the reduction in the DPPH^•^ radical by the addition of a proton atom from a food component that exhibits antioxidant activity and its conversion into the hydrazine DPPH-H. When a substance that exhibits antioxidant activity is added to the deep purple methanolic DPPH^•^ solution, the free radical is reduced to DPPH-H, which possesses a yellow color, thus resulting in the reduction in optical density.

### 4.2. ABTS^•+^ Radical Scavenging Assay

The proposed ABTS^•+^ radical scavenging assay was described by Cano et al. [33] and has a similar mechanistic principle with the DPPH^•^ radical scavenging assay. To be more specific, the addition of an electron or a hydrogen proton can neutralize the ABTS^•+^ radical as follows:ABTS^•+^ _(green at 730 nm)_ + ArOH → ABTS(H) _(colorless)_ + ArO^•^

One of the main differences as compared to the DPPH^•^ radical, which already exists as a stable free radical, is that the ABTS^•+^ radical is produced by the oxidation of ABTS. Chemical reactions with oxidizing agents or enzymes, such as peroxidases, are responsible for the oxidation of ABTS reagent [34]. This experimental setup prevents from potential off-site interactions between the oxidizing agents used for the oxidation of ABTS and the antioxidant compounds of food products. Following the formation of ABTS^•+^ radical in the presence of H_2_O_2_ catalyzed by the enzymatic activity of horseradish peroxidase, it reacts with hydrophilic and lipophilic antioxidant molecules that are present in food products [32,35]. The ABTS^•+^ radical remains stable after its formation, a critical advantage for the outcome of this assay. This radical has a green color and its optical density can be measured at 730 nm. The addition of a sample containing antioxidant molecules to the aqueous ABTS^•+^ radical solution leads to decolorization and to the reduction in optical density.

As previously mentioned, the DPPH^•^ and ABTS^•+^ radical scavenging assays can identify the presence of lipophilic and both lipophilic and hydrophilic antioxidants, respectively [36]. Therefore, their simultaneous application can detect the total of the antioxidant compounds contained in food products.

### 4.3. OH^•^ Radical Scavenging Assay

OH^•^ radicals are strong oxidizing agents, considered as the most reactive form of ROS. They are critical inducers of cellular damage as they oxidize DNA and RNA, leading to cytotoxicity, mutagenesis, and carcinogenesis [37]. The ability of an extract to scavenge the naturally occurring OH^•^ radicals is directly related to its antioxidant capacity. The proposed OH^•^ radical scavenging assay was firstly described by Osawa and Kawakishi [38] in order to assess the antioxidant efficacy against these endogenous prooxidants. The method relies on the oxidation of 2-deoxyribose. During the Fenton reaction, OH^•^ radicals oxidize 2-deoxyribose into malondialdehyde (MDA), leading to a colorimetric outcome that can be measured spectrophotometrically at 520 nm:2-deoxyribose + OH^•^ → MDA

The ability of an extract to scavenge OH^•^ radicals is estimated as the rate of inhibition of 2-deoxyribose oxidation.

### 4.4. O_2_^•−^ Radical Scavenging Assay

The naturally occurring O_2_^•−^ radicals can cause enzyme inactivation, cell membrane degradation, and cell death. They are responsible for the peroxidation of polyunsaturated fatty acids [39], while MDA and 4-hydroxynonal (4-HNE), both of them byproducts of lipid peroxidation, are mutagenic and carcinogenic [40]. Furthermore, O_2_^•−^ radicals can damage DNA at guanine residues, causing mutations that can result in cancer [41]. The proposed O_2_^•−^ radical scavenging assay was described by Gülçin et al. [42]. The O_2_^•−^ radicals are derived from the phenazine methosulfate (PMS)–nicotinamine adenine dinucleotide hydride (NADH) system through the oxidation of NADH, and can be measured through the reduction in nitro blue tetrazolium (NBT) at 560 nm:PMS-NADH + O_2_ → O_2_^•−^; O_2_^•−^ + NBT^2+^ (yellow) → formazan/NBT (blue)

Substances with antioxidant capacity are able to inhibit NBT formation.

### 4.5. Alkaline DMSO Assay

The alkaline dimethylsulfoxide (DMSO) assay is another antioxidant method that can be used to evaluate the O_2_^•−^ radical scavenging capacity of a sample [43]. To be more specific, O_2_^•−^ radicals are formed by adding sodium hydroxide (NaOH) to air-saturated DMSO [44]. At room temperature, the generated O_2_^•−^ radicals are stable and convert NBT to formazan, a reaction that can be determined spectrophotometrically at 560 nm [45]. The reduction in NBT is determined by the presence and absence of the test sample.

It is worth mentioning that a modified version of alkaline DMSO has been proposed for the purpose of overcoming pitfalls and shortcomings of the original assay, such as the interference of reducing compounds with the probe molecule, i.e., NBT [46]. Based on this, the probe molecule is substituted by *N*,*N*-dimethyl-*p*-phenylenediamine dihydrochloride (DMPD), a compound reduced by O_2_^•−^ radicals, and not by other reducing agents [46].

### 4.6. Reducing Power Assay

Metal ions are a double-edged sword for biological systems; they have substantial roles in various physiological processes [47]; however, they can contribute to the generation of ROS through the Fenton and Haber–Weiss reactions. The reducing power of a substance is a measure of its electron-donating ability, and as a result, it constitutes a reliable indicator of antioxidant capacity. The proposed experimental protocol was firstly described by Yen and Duh [48] in order to determine the ability of a sample to reduce ferric ions (Fe^+3^) to ferrous ions (Fe^+2^), as follows:2Fe(CN)_6_^3−^ + antioxidant − H → 2Fe(CN)_6_^4−^ + antioxidant^−^ + H+

This reduction leads to a reaction with ferrocyanide, forming a complex that can be measured spectrophotometrically at 700 nm. The initial yellow color of the sample changes to different shades of green and blue, depending on the reducing power efficacy:4Fe^3+^ + 3Fe(CN)_6_^4−^ (yellow) → Fe_4_[Fe(CN)_6_]^3^ (Prussian blue)

The higher the absorption, the greater the reducing power. Compounds with a reducing power potential are strong electron donors and can reduce oxidized intermediates, acting as primary or secondary antioxidant compounds.

### 4.7. FRAP Assay

The FRAP assay represents another iron (III)-based antioxidant method, which can be applied for assessing the antioxidant capacity of a sample based on its ferric reducing ability. This method was initially introduced by Benzie and Strain [49] to investigate the antioxidant power of human plasma; however, in recent years, its use has been extended to the determination of the antioxidant activities of various food products [50,51]. More elaborately, it evaluates the ability of antioxidant compounds to reduce Fe^+3^ ions, forming a complex with 2,4,6-tripyridyl-S-triazine (TPTZ), to the more stable Fe^2+^ ions via a single electron transfer (SET) at acidic pH [52], as follows:[Fe^III^(TPZT)_2_]^3+^ + ArOH → [Fe^II^(TPZT)_2_]^2+^ + ArO^•^ + H^+^

The reduction in Fe^3+^ ions to Fe^2+^ ions leads to the generation of an intense violet-blue color, and the change in optical density can be measured spectrophotometrically at 593 nm.

### 4.8. Plasmid DNA Relaxation Assay

The ability of food products to protect against the free radical-induced DNA damage can be evaluated using the plasmid DNA relaxation assay [53,54]. This method has been previously described by Priftis et al. [55] to assess the protective actions of foodstuffs against DNA single-strand breaks induced by peroxyl radicals (ROO^•^). The thermal decomposition of 2,2′-azobis(2-amidinopropane hydrochloride) (AAPH) leads to ROO^•^ generation, as follows:R − N = N − R → 2R^•^ + N_2_; R^•^ + O_2_ → ROO^•^

Plasmid DNA exists natively in its supercoiled conformation [56]; the addition of an azo compound, such as AAPH, to the reaction and its thermal decomposition lead to the generation of ROO^•^ and to the formation of single-strand breaks. In this way, the plasmid DNA is converted to its open circular conformation, which runs slower than the supercoiled when electrophoresed on an agarose gel. However, in the presence of an antioxidant compound, ROO^•^ are scavenged and the plasmid DNA retains its supercoiled conformation.

### 4.9. TBARS Assay

The ability of food products to inhibit lipid peroxidation can be evaluated using the TBARS assay, described previously by Dissanayake et al. [57]. According to the method, the rich-in-lipids egg yolk is used as the lipid substrate. Linoleic acid and linolenic acid are two of the most important polyunsaturated fatty acids present in this substrate. When these fatty acids react with oxygen, they produce MDA, which then reacts with thiobarbituric acid (TBA), to produce a pink-colored product that absorbs maximum at 532 nm. In the presence of an antioxidant compound, the oxidized lipids are reduced, resulting in a decrease in optical density.

### 4.10. CUPRAC Assay

The ability of food products to reduce cupric ions (Cu^2+^) to cuprous ions (Cu^1+^) in the presence of a chelating agent can be evaluated using the CUPRAC assay described by Trofin et al. [58]. Neocuproine, bathocuproine, and bathocuproinedisulfonic acid disodium salt are chelating agents used for this purpose. The CUPRAC assay is based on the reduction in Cu^2+^ into Cu^1+^ by the action of the antioxidant compounds present in the tested sample. In the presence of an antioxidant compound, the CUPRAC reagent (CuCl_2_ + Nc + NH_4_Ac) is reduced to a colored product that can be measured spectrophotometrically at 450 nm.

## 5. Agri-Food Products Examined via In Vitro Antioxidant Cell-Free Assays

### 5.1. Dairy Products

Dairy products contain valuable nutrients and antioxidant compounds, both lipophilic and hydrophilic, in varying amounts, depending on the matrix type (milk, yogurt, fermented milk, cheese) and the processing method [59]. Their antioxidant capacity is primarily related to the content of antioxidant components, which are high in sulfur amino acids and vitamins A, E, and C or carotene. However, previous studies have demonstrated that their antioxidant properties are also associated with the interactions between the phenolic compounds of milk and milk proteins [60,61]. Biopeptides produced during cheese fermentation or maturation also exhibit antioxidant activities [62]. For this reason, cheeses have the highest antioxidant potential among dairy products, owning to their higher protein content and fermentation process [59].

To determine the antioxidant capacity of dairy products, several methods have been applied. More specifically, milk and fermented milk samples, have been evaluated for their free radical scavenging capacity against the DPPH^•^ radical [60,63]. The comparison of the antioxidant properties of different types of milk, including infant formula, yogurt, fresh cream cheese, and kefir using the specific assay has demonstrated that the strongest antioxidant activities are exerted by the regular whole ultra-high-temperature (UHT) milk [64]. Furthermore, the assessment of the antioxidant capacity of milk using the CUPRAC assay and ABTS^•+^ radical scavenging assay can reveal the contribution of proteins to the antioxidant properties. According to these assays, milk with a higher content of fatty acids exhibit the strongest antioxidant activities [65].

The antioxidant activity of kefir, a fermented milk drink, may be attributed to its proton-donating abilities and reducing properties, as evidenced by the DPPH^•^ and O_2_^•−^ radical scavenging assays [66]. Gupta et al. have investigated the effects of cheese maturation in different cheddar cheeses, using a panel of antioxidant assays. An evaluation of the O_2_^•−^ radical scavenging capacity of water-soluble extracts of various stages of cheese maturation has revealed that the antioxidant properties increase during maturation [67]. Papadaki and Roussis have investigated the antioxidant potency of different Greek yogurts, using a variety of in vitro cell-free assays, concluding that the water-soluble constituents of cow yogurt with a lower fat content exert higher ROS scavenging activities, as compared to those from cow yogurt with a higher fat content [68].

### 5.2. Honey Products

The honey bee, *Apis mellifera,* produces honey and honey-related products, such as propolis and beeswax [69]. Honey is a complex mixture of compounds, particularly rich in sugars and, in lower concentrations, in vitamins, amino acids, and polyphenols [70,71]. According to the scientific literature, the latter compounds render honey a bio-functional food product with strong antibacterial [72], anti-inflammatory [73], and antioxidant [74,75] properties. The floral source, the location of the beehive, and the climate and soil conditions are significant parameters that affect the composition of honey in antioxidant compounds [76,77,78]. Herein, we mention previous studies that have examined the antioxidant properties of honey and honey-related products, using in vitro cell-free techniques.

A previous study by our research group assessed the antioxidant capacity of 21 honey types, produced in Mount Olympus, Greece, using the DPPH^•^ and ABTS^•+^ radical scavenging assays. Our results showed that a multifloral honey, comprising mint, herbs, and acacia, possessed the highest polyphenolic content and exerted the strongest antioxidant activities, a finding supported by the lowest IC_50_ values in DPPH^•^ radical scavenging assay and ABTS^•+^ radical scavenging assay, i.e., 7.5 mg/mL and 4.5 mg/mL, respectively [79]. Furthermore, in a recent study by our research team, we evaluated the antioxidant potency of six honey samples, produced in Mount Taygetos and in Mount Pindos, Greece, using a complete set of in vitro cell-free screening techniques, as well as in vitro cell-based systems. According to our results, the forest honey with oak honeydew exhibited the strongest antioxidant activities in most of the cell-free antioxidant assays examined. Finally, an important finding of the study was that most of the examined honey samples exerted detrimental effects on the redox homeostasis of HepG2 cancer cell line by promoting lipid peroxidation and protein carbonylation [19,79].

Nagai et al. have evaluated the free radical scavenging capacity of six monofloral honey samples against the endogenous OH^•^ and O_2_^•−^ free radicals in order to identify potential changes in antioxidant potency attributed to the temperature increase [80]. Moreover, Almeida et al. have applied the CUPRAC assay and the reducing power assay in combination with other methods (moisture content, diastase activity, etc.) to categorize 15 honey samples through the identification of their intrinsic characteristics [81]. Tahirovic et al. have proposed a combination of ROO^•^ and OH^•^ radical scavenging assays in order to categorize honey samples on the basis of their antioxidant capacity, concluding that forest honeys possess the strongest antioxidant properties [78].

### 5.3. Medicinal Plants and Herbs

Medicinal plants and herbs are particularly rich in natural bioactive compounds. The phenolic compounds, which are plant secondary metabolites, exert potent antimicrobial, antimutagenic, and anti-inflammatory activities [82,83]. The antioxidant properties of plant polyphenols represent a major area of research interest, as they can act as hydrogen and electron donors due to their chemical structure, containing several hydroxyl groups on aromatic rings [83].

In an effort to investigate the antioxidant capacity of medicinal plants and herbs, a series of in vitro cell-free assays have been applied. A previous study by our research team investigated the antioxidant properties of a large number of herb decoction extracts from the Epirus region, Greece, in in vitro cell-free systems. More elaborately, we examined their free radical scavenging capacity against the DPPH^•^, ABTS^•+^, and O_2_^•−^ free radicals, the reducing properties through the reducing power assay, and the antigenotoxic properties through the plasmid DNA relaxation assay. Our results demonstrated that different herb decoction extracts were the more efficacious in each antioxidant assay. However, with regard to DPPH^•^ and ABTS^•+^ radicals, an *Origanum vulgare* herb decoction extract showed the strongest scavenging activity at concentrations < 20 µg/mL [20]. In line with this finding, another study has indicated that the oregano essential oils exhibit a higher antioxidant capacity against the DPPH^•^ and ABTS^•+^ radicals as compared to rosemary, with IC_50_ values of 1.39 and 2.46 mg/mL, respectively [84]. Contrariwise, a recent research study has demonstrated that oregano oils exhibit the highest ABTS^•+^ scavenging activity, as evidenced by the IC_50_ value, i.e., 0.08 g/L; however, they do not show the same efficacy in CUPRAC assay, wherein thyme oils display stronger reducing properties [85].

A finding of particular interest is that the extraction method and the solvents used have a significant impact on the biological activity of plant extracts. More elaborately, the hydroalcoholic extract of *Echium amoenum* exerts a stronger scavenging activity against OH^•^ radicals (110.8 μg/mL), as compared to the infusion and decoction extracts (124.1 and 129.1 μg/mL, respectively) [86]. Nanda et al. have examined the polyphenolic content and the reducing properties of five different medicinal herbs, concluding that the aqueous extract of *Ocimum basilicum* is the richest in phenolic and flavonoid content, as well as the most efficient reducing agent. Furthermore, this aqueous extract exerts the most potent free radical scavenging activities and inhibits lipid peroxidation, as assessed via the TBARS assay using the egg yolk as a lipid substrate [87]. Finally, Chaves et al. evaluated the antioxidant properties of 12 species of Mediterranean plant extracts using the DPPH^•^ and ABTS^•+^ radical scavenging assays and the reducing power assay; however, their results showed differences in sensitivity. To be more specific, ABTS^•+^ radical scavenging assay shows a lower antioxidant activity, whereas the reducing power assay is the most sensitive and, as a result, is the one establishing more differences between the examined plant species [88].

### 5.4. Olive Products

Olive cultivation (*Olea europaea* L.) is an integral part of the agricultural economics of Mediterranean countries. Olive oil and table olives are the main olive products with commercial interest and are the foundation of the Mediterranean diet. The cultivation of olive trees and the process of olive oil production is accompanied by the generation of large amounts of olive by-products, annually, with high pollution load, including olive leaves, branches, olive brines, and olive pomace [89].

In recent years, there has been an increasing interest in the food industry for the discovery and exploitation of plants and/or their by-products as a natural source of antioxidants [90,91,92,93]. Therefore, the chemical composition of olive products and by-products is of particular research interest, constituting rich sources of nutrients and bio-functional components, such as phenolic compounds [94]. As stated above, phenolic compounds are an essential group of secondary metabolites that are produced from plants in response to abiotic and biotic stressful factors [95,96,97]. Previous studies have clearly demonstrated that olive products and by-products exert antioxidant, anti-inflammatory, anti-microbial, and anticancer properties mainly due to their high content in phenolic compounds. In general, the main categories of phenolic compounds present in olive products and by-products are simple phenols and acids, lignans, secoiridoids, and flavonoids [97,98].

The phenolic content of olive extracts depends on several factors, including the extraction procedure and the solvent used [99], the storage conditions [100,101], the part of the tree from which the extract is derived [70,102], and the cultivar itself [103]. In order to determine the in vitro antioxidant activity of olive derivative extracts [104,105,106,107] or of pure isolated bioactive molecules, such as oleuropein, hydroxytyrosol, and tyrosol [108,109,110], a wide variety of laboratory assays are used. Indisputably, the DPPH^•^ and ABTS^•+^ radical scavenging assays are the most common methods applied for the determination of the free radical scavenging efficacy. The OH^•^ and O_2_^•−^ radical scavenging assays are also used due to the biological relevance of these naturally occurring free radicals that can cause severe oxidative damage to cellular components. Furthermore, the antioxidant activity of olive extracts is evaluated by determining their reducing properties with the ferric reducing antioxidant power (FRAP) assay and the CUPRAC assay [111,112,113,114].

With regard to the antioxidant activities, experimental data from previous studies have revealed that extracts derived from the same part of the olive tree, although extracted using different solvents, exhibit different IC_50_ values [115,116,117]. This phenomenon is attributed to the extraction of non-polar or polar components, which eventually determines the antioxidant capacity of the extract.

A previous study of our research group has investigated the antioxidant properties of an olive oil total polyphenolic fraction and of purified hydroxytyrosol in in vitro cell-free and cell-based systems. According to our results, both the total polyphenolic fraction and the pure hydroxytyrosol exert potent free radical scavenging efficacies against the DPPH^•^ and ABTS^•+^ radicals, also improving the redox status of the endothelial cells and myoblasts by enhancing their antioxidant defenses [118]. Hydroxytyrosol is a potent antioxidant molecule and its content in olive oil has a significant impact on the antioxidant effects. More specifically, a previous study by our research team has examined the antioxidant properties of olive oil extracts with significant differences in their hydroxytyrosol and tyrosol content in cell-free and cell-based systems. According to our findings, the polyphenolic extracts with high hydroxytryrosol content are more efficacious in terms of their antioxidant and antigenotoxic activities with regard to the extracts with a higher tyrosol content [119].

By-products of table olive preparation and of olive oil production are rich in antioxidant compounds and can be utilized in the food industry as natural antioxidants, feed additives, and components for the production of bio-functional food products. For instance, the enrichment of animal feed with a high phenolic feed material derived from olive mill wastewater (OMW) enhances the animal redox status [120,121,122]. Furthermore, polyphenolic extracts derived from olive tree blossoms exert strong antioxidant, antigenotoxic, and antimutagenic activities, also improving the redox status of endothelial, cervical, and liver cells, and myoblasts by enhancing the antioxidant defense mechanisms and by decreasing ROS levels [123]. Finally, a recent study of our research group has evaluated the antioxidant and antigenotoxic properties of a brine extract of Kalamon olives debittering, using both in vitro cell-free and cell-based assays. According to the experimental protocol, the free radical scavenging efficacies against the DPPH^•^, ABTS^•^, and O_2_^•−^ radicals, the protective ability against the ROO^•^-induced DNA damage, and the reducing properties are all examined. Our results confirm that the brine extract of Kalamon olives debittering exhibits strong antiradical, antigenotoxic, and reducing activities in cell-free systems [124].

### 5.5. Fruits and Vegetables

Fruit and vegetable intake in daily nutrition is associated with anti-inflammatory, anti-diabetic, anti-cancer, and neuroprotective properties. Fruits and vegetables are particularly rich in bioactive compounds with antioxidant properties that reduce the oxidative stress, thus protecting the cardiovascular and nervous system. More specifically, they contain a complex mixture of antioxidant compounds, such as polyphenols, vitamins A, B, C, and E, carotenoids, and fibers. Phenolic acids that are present in fruits and vegetables, including gallic acid, syringic acid, coumaric acid, and ferulic acid, protect biomolecules from oxidative damage [125].

Previous studies have examined the ability of fruit and vegetable extracts to scavenge free radicals or to possess reducing properties, acting as donors of hydrogen atoms or electrons to oxidized intermediates. The most widely applied methods to estimate the free radical scavenging capacity of such extracts are the DPPH^•^ and ABTS^•+^ radical scavenging assays. The high total phenolic and anthocyanin content of fruit extracts is associated with a high antioxidant capacity [126]. It is worth noting that the antioxidant composition of fruits and vegetables is affected by various factors, such as the climate, the soil geochemistry, and the cultivation practices. Apostolou et al. have demonstrated that the antioxidant capacity of grape stem extracts differs significantly between two consecutive harvest years, probably due to the different climatic conditions [21]. Furthermore, the ability of cherry fruits to neutralize DPPH^•^ radicals is significantly affected by the soil properties, while the elevation of the cultivars is a key determinant for the polyphenolic content of cherry fruits [127].

## 6. Discussion

The use of the proposed antioxidant assays might play a crucial role in understanding the endogenous health-promoting properties of agri-food products. Over the last several decades, the scientific community has been persistently proposing for an investigation of mechanisms and tools that will enlighten the oxidative processes occurring in foods, and at the same time, their beneficial and health-promoting properties. Therefore, research on the improvement of human wellness and product quality based on scientific criteria established in each food category has been a perpetual debate for scientists seeking to find reproducible, cost-effective, and simple assays for classifying food bioactivity.

In this debate, the main task needed to fulfill through the selected assays is the ability to “compare food choices”. The measurement of foods’ antioxidant capacity might enable comparisons between different food choices and varieties. It helps to determine which foods are rich in antioxidants and may have higher potential for providing health benefits [128] through the scavenging of free radicals and other mechanisms that lead to protection against oxidative damage [129]. Therefore, researchers will be able to define the nutritional content of every individual product and henceforward identify the potential health benefits associated with consuming those foods. This information can guide individuals in making informed dietary choices to incorporate antioxidant-rich foods into their meals; in case needed.

Apart from the resulting benefits for consumers, the food industry is another pillar affected. Establishing experimental tools, as the network of assays described herein, comprises a crucial mission for quality control and product development purposes in the food industry [16]. Food manufacturers can ensure that their products meet certain measurable standards and can be placed in label claims [16]. Additionally, measuring antioxidants can aid in the development of new food products or formulations with enhanced antioxidant properties [16]. Furthermore, consumers might entrust product labels and can be assured that they are receiving the expected quality food product based on bioactivity measurements [16].

Antioxidant measurements in foods are also vital for scientific research and health studies, enabling the scientific community to be one step closer in the investigation of the relationship between dietary antioxidants and various health outcomes. Measuring antioxidants allows for the identification of potential associations between antioxidant intake and a reduced risk of chronic diseases such as cardiovascular diseases, cancer, and neurodegenerative disorders [130] contributing to the growing body of knowledge on nutrition and health. Specifically, a meta-analysis of the data extracted by these assays might be the basis for establishing dietary recommendations and guidelines in several pathologies [131,132]. In the previous section, we only mention some food categories that have been evaluated with the proposed assays. National and international health organizations can use such scientific evidence on antioxidant content to formulate guidelines on recommended daily intakes of antioxidants. These recommendations will help individuals make dietary choices that support optimal health and well-being.

## 7. Conclusions

The consumption of food products containing high amounts of bioactive compounds with antioxidant properties could be beneficial for human health and contribute to the prevention of chronic diseases, such as cardiovascular diseases, neurodegenerative disorders, and cancer. Hence, there is a growing interest in the food industry for the development of methodologies that can identify the beneficial health effects of food products. The establishment of reliable and valid experimental protocols that can evaluate the antioxidant capacity of food products has received considerable interest due to the fact that the adoption of a healthy diet protects from malnutrition and decreases the risk of developing chronic diseases. Toward this purpose, the present article proposed a novel approach to investigate the antioxidant potency of various food products based on a panel of well-established in vitro cell-free assays. These antioxidant assays meet the strict criteria mentioned above and are rapid, cost-effective, and reproducible among different food samples. The adoption of a panel of assays that investigate multiple properties of agri-food products will contribute to both monitoring their quality based on these biological criteria and to a better communication of the existing foods classification to consumers. In summary, measuring antioxidants in foods is crucial for understanding their nutritional content, antioxidant capacity, and potential health benefits. It guides food choices, aids in quality control, supports research efforts, and informs dietary recommendations and guidelines.

## Figures and Tables

**Table 1 ijms-24-16447-t001:** Summary table demonstrating the antioxidant properties of various natural products evaluated using in vitro cell-free screening techniques. All results in [17,18,21,22] are expressed as mean ± standard deviation (SD). All results in [19,20] are expressed as mean ± standard error of the mean (SEM). IC_50_ (Half maximal inhibitory concentration): The concentration of the sample required for the inhibition of the 50% of the corresponding free radicals. AU_0.5_ (Absorbance unit 0.5): The concentration of the tested sample required for the achievement of an absorbance value of 0.5. * Refers to the concentration of each sample that has the ability to scavenge 20% of the free radical (IC_20_).

Samples	Antioxidant Assays						References
	ABTS^•+^ Scavenging Assay	DPPH^•^ Scavenging Assay	O_2_^•−^ Scavenging Assay	OH^•^ Scavenging Assay	Reducing Power Assay	Plasmid DNA Relaxation Assay	
Wine extracts	IC_50_ (μg/mL)	IC_50_ (μg/mL)	IC_50_ (μg/mL)	IC_50_ (μg/mL)	AU_0.5_ (μg/mL)	IC_50_ or IC_20_ * (μg/mL)	
Xinomavro	7.3 ± 0.19	13.4 ± 0.42	34.5 ± 3.17	304.8 ± 29.57	4.9 ± 0.07	260.5 ± 27.4	[17]
Agiorgitiko	8.2 ± 0.04	14.5 ± 0.62	32.0 ± 0.37	491.2 ± 30	8.3 ± 0.59	116.1 ± 19.4	
Assyrtiko	18.4 ± 1.05	28.4 ± 2.27	73.9 ± 0.75	165.7 ± 13.03	13.0 ± 0.21	220.3 ± 14.1 *	
Malagouzia	43.5 ± 1.33	89.4 ± 4.14	268.5 ± 33.62	409.1 ± 19.03	48.1 ± 0.66	150.1 ± 15.0 *	
Honey	IC_50_ (mg/mL)	IC_50_ (mg/mL)	IC_50_ (mg/mL)	IC_50_ (mg/mL)	AU_0.5_ (mg/mL)	IC_50_ (mg/mL)	
Oak	2.96 ± 0.81	7.14 ± 0.02	1.98 ± 0.04	1.22 ± 0.04	1.87 ± 0.19	2.98 ± 0.11	[19]
Eryngium creticum	4.03 ± 0.08	9.95 ± 0.025	7.48 ± 0.37	1.04 ± 0.06	3.60 ± 0.3	6.04 ± 0.19	
Fir and vanilla	1.03 ± 0.01	6.51 ± 0.32	1.01 ± 0.01	1.05 ± 0.06	2.41 ± 0.01	1.60 ± 0.17	
Forest with oak honeydew	0.90 ± 0.01	4.61 ± 0.29	1.24 ± 0.01	1.24 ± 0.02	1.79 ± 0.06	1.55 ± 0.15	
Flower (1)	1.99 ± 0.1	15.04 ± 0.3	4.32 ± 0.14	0.68 ± 0.01	3.71 ± 0.25	9.02 ± 0.41	
Flower (2)	1.45 ± 0.02	8.47 ± 0.69	2.63 ± 0.02	0.66 ± 0.01	2.28 ± 0.01	6.86 ± 0.68	
Herb extracts	IC_50_ (μg/mL)	IC_50_ (μg/mL)	IC_50_ (μg/mL)	IC_50_ (μg/mL)	AU_0.5_ (μg/mL)	IC_50_ (μg/mL)	
*Origanum vulgare*	7.85 ± 0.56	6.60 ± 1.50	12 ± 1.07	-	7.5 ± 0.41	35 ± 3.06	[20]
*Salvia officinalis*	19.07 ± 0.09	26.68 ± 1.22	6.5 ± 0.25	-	8 ± 0.35	54 ± 4.51	
*Aloysia citrodora*	8.29 ± 1.13	10.25 ± 0.25	49 ± 2.39	-	3.5 ± 0.13	26 ± 2.14	
*Rosmarinus officinalis*	12.27 ± 0.38	11.63 ± 4.32	14.5 ± 1.09	-	7.5 ± 0.48	25 ± 1.27	
Fruit extracts	IC_50_ (mg/mL)	IC_50_ (mg/mL)	IC_50_ (mg/mL)	IC_50_ (mg/mL)	AU_0.5_ (mg/mL)	IC_50_ (mg/mL)	
*Lycium barbarum*	0.67 ± 0.01	2.33 ± 0.03	-	-	-	1.80 ± 0.05	[18]
Grape seed extracts	IC_50_ (μg/mL)	IC_50_ (μg/mL)	IC_50_ (μg/mL)	IC_50_ (μg/mL)	AU_0.5_ (μg/mL)	IC_50_ (μg/mL)	
*Mavrotragano*	5.0 ± 0.4	3.5 ± 0.3	-	400 ± 55	-	0.65 ± 0.07	[21]
*Voidomato*	7 ± 0.9	3.5 ± 0.5	-	200 ± 24	-	1 ± 0.08	
*Moshato*	8 ± 0.7	3.5 ± 0.5	-	400 ± 39	-	1 ± 0.1	
*Vinsanto*	10 ± 1.1	4 ± 0.2	-	300 ± 38	-	0.95 ± 0.08	
*Athiri*	15.0 ± 1.6	5.0 ± 0.4	-	310 ± 25	-	1.05 ± 0.07	
*Mandilaria*	9.0 ± 1.0	9.0 ± 0.9	-	390 ± 35	-	1.05± 0.12	
Plant extracts	IC_50_ (μg/mL)	IC_50_ (μg/mL)	IC_50_ (μg/mL)	IC_50_ (μg/mL)	AU_0.5_ (μg/mL)	IC_50_ (μg/mL)	
*Mentha microphylla*	29 ± 1.2	15 ± 0.6	-	240 ± 43	-	0.5 ± 0.09	[22]
*Mentha longifolia*	28 ± 1.0	36 ± 1.2	-	325 ± 24	-	1.55 ± 0.18	
*Sideritis raeseri ssp. raeseri*	31 ± 0.6	38 ± 1.5	-	>800	-	2.20 ± 0.06	
*Salvia pomiferassp. calycina*	19 ± 1.0	19 ± 0.6	-	170 ± 13	-	1.25 ± 0.10	
*Salvia fruticosa*	16 ± 0.5	29 ± 0.6	-	350 ± 14	-	0.95 ± 0.08	
*Salvia sclarea*	25 ± 0.5	20 ± 1.2	-	210 ± 4	-	1.10 ± 0.18	
*Salvia officinallis*	21 ± 1.0	17 ± 0.6	-	300 ± 14	-	0.90 ± 0.07	

## Data Availability

Not applicable.

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
