# Peer review of "Development of a Holistic In Vitro Cell-Free Approach to Determine the Redox Bioactivity of Agricultural Products"

_ijms, 2023, doi:10.3390/ijms242216447_

Round 1

Reviewer 1 Report

Comments and Suggestions for Authors

Due to the fact that in recent years there has been a high consumer demand for food products that provide nutritional benefits for human health, I believe that the topic is up-to-date and fits into the current trends on the food products market.

This paper presents a novel methodology incorporating a full suite of in vitro cell-free screening techniques to assess the bioactivity of various food products based on their antioxidant capacity. Taking up this topic is justified and contains elements of novelty.

Comments:

- I propose in the article a short discussion on the applicable legal regulations regarding food products and bodies certifying food products

- in point 2 regarding the assessment of the quality of food products, it should be mentioned specifically what tests are performed, apart from testing the biological properties of a wide range of lipophilic and hydrophilic antioxidant substances

Author Response

Dear reviewer,

Please find attached our responses to your comments.

Thank you!

Reviewer 2 Report

Comments and Suggestions for Authors

The review is very well written with minor corrections required.

-Line 92 21st should be superscript

- Line 146 and all after IC50 the 50 should be subscript

Whilst a good overall review of the different methodologies have been provided with regards to Ferric reducing capacity the use of  2,4,6-trypyridyl-s-triazine (TPTZ) has not been included.

Similarly with respect to superoxide assays these have not been fully discussed especially in the presence of reducing compounds which could potential interfere with the probe molecule such as in the case of the Alkaline DMSO (Dimethyl Sulfoxide) Assay which could be modified using DMPD  rather than NBT.

Missing completely from the review is the structural relationship between the observed antioxidant activity. It is suggested that the authors include the Bors criteria.

Comments on the Quality of English Language

The review is very well written with minor corrections required.

-Line 92 21st should be superscript

- Line 146 and all after IC50 the 50 should be subscript

Whilst a good overall review of the different methodologies have been provided with regards to Ferric reducing capacity the use of  2,4,6-trypyridyl-s-triazine (TPTZ) has not been included.

Similarly with respect to superoxide assays these have not been fully discussed especially in the presence of reducing compounds which could potential interfere with the probe molecule such as in the case of the Alkaline DMSO (Dimethyl Sulfoxide) Assay which could be modified using DMPD  rather than NBT.

Missing completely from the review is the structural relationship between the observed antioxidant activity. It is suggested that the authors include the Bors criteria.

Author Response

(The authors gave the same response as above.)
